# Factors Influencing the Bioavailability of Organic Molecules to Bacterial Cells—A Mini-Review

**DOI:** 10.3390/molecules27196579

**Published:** 2022-10-04

**Authors:** Wojciech Smułek, Ewa Kaczorek

**Affiliations:** Institute of Chemical Technology and Engineering, Poznan University of Technology, Berdychowo 4, 60-965 Poznan, Poland

**Keywords:** bacteria, availability, membranes, organic chemicals, solubility

## Abstract

The bioavailability of organic compounds to bacterial cells is crucial for their vital activities. This includes both compounds that are desirable to the cells (e.g., sources of energy, carbon, nitrogen, and other nutrients) and undesirable compounds that are toxic to the cells. For this reason, bioavailability is an issue of great importance in many areas of human activity that are related to bacteria, e.g., biotechnological production, bioremediation of organic pollutants, and the use of antibiotics. This article proposes a classification of factors determining bioavailability, dividing them into factors at the physicochemical level (i.e., those related to the solubility of a chemical compound and its transport in aqueous solution) and factors at the microbiological level (i.e., those related to adsorption on the cell surface and those related to transport into the cell). Awareness of the importance of and the mechanisms governing each of the factors described allows their use to change bioavailability in the desired direction.

## 1. Introduction

Every living cell—including bacterial cells—is a distinct system, separated from the external environment, capable of independent development and replication associated with the transfer of individual characteristics through genetic material. Nevertheless, an important feature of the functioning of living cells is their interaction with the surrounding environment, which manifests itself mainly through a mass exchange, i.e., the transport of chemical compounds into and out of the cell [1,2].

In the case of small-molecule compounds (such as water molecules, gases, etc.) and inorganic cations and anions, their exchange between the cell and the environment (usually an aqueous solution, even if based only on the water contained in the biofilm, for example) is relatively intensive [3]. These are molecules present in relatively high concentrations, while also being very mobile and easily diffusing in the aqueous environment. Moreover, their permeation through the cell wall and membrane occurs both passively (e.g., by osmosis) and actively (e.g., by transport proteins) [4,5].

Compared to small-molecule compounds, organic compounds taken up by cells have a significantly higher molecular weight, which usually ranges from tens to thousands of Da (bigger molecules, such as some biopolymers, often need to be lysed before they can be transferred into the cell) [6]. In their case, active transport via the cell membrane is preferred [3]. As a result, their behaviour in aqueous solution and permeation through the cell wall and membrane differs significantly from that observed for the aforementioned small-molecule compounds [7].

## 2. Challenges in Defining Microbial Bioavailability

The extent to which chemical compounds are realistically obtainable by the cell is referred to as bioavailability. This very general definition has already been made more precise and specific on several occasions.

Tardif and Brodeur [8] described bioavailability as the percentage of an administered dose of a xenobiotic that reaches the systemic circulation of a living organism. Analogically, this term is defined by Wolverton [9]. Similarly, Willhite et al. [10] stated that bioavailability is “the extent to which a material is taken up into a living organism exposed to that substance (absorbed dose)”. In environmental sciences, bioavailability is defined as the fraction of the total mass of a compound present in a compartment that has the potential to be absorbed by the organism [11]. However, it should be noted that in most cases bioavailability has been defined in the context of the transport of organic compounds into the cells and tissues of higher organisms, e.g., the pharmacokinetics of drugs or the action of herbicides on plants. This approach involves some narrowing of the definition, due to the specificity of the cells of higher organisms and the differences between them and the cells of microorganisms [12].

Moreover, other similar terms have also been introduced to describe the interactions between living (micro)organisms and chemical compounds. Kramer and Ryan [13] applied a quite different term, i.e., bioaccessibility, to refer to the total amount of a contaminant that is desorbed from the soil and available for uptake into the circulatory system. Ortega-Calvo et al. [13], in their review article (based on studies of Ehlers and Luthy [14], Semple et al. [12], and Reichenberg and Mayer [15], among others), concluded that the term “bioavailability” frequently focuses on the aqueous or dissolved contaminant, the term “bioaccessibility” relates to the incorporation of the rapidly desorbing contaminant during the exposure, and the “chemical activity” determines the potential biological effects of the dissolved contaminant (Figure 1). 

The term “bacterial bioavailability” was defined, probably for the first time, by Ropponen et al. [16], who stated that the term should refer to the rate and extent to which a drug is available at the target site. Considering antibiotic therapy, bioavailability is usually determined by absorption, distribution, metabolism, and excretion from the treated organism, but these mechanisms can also be applied to the compartment of bacterial cells. Moreover, Ropponen et al. [16] noted that the distribution of an antibiotic between different bacterial compartments may be of importance—especially in Gram-negative bacteria, because of the complexity of their cell wall, which includes two membranes.

The difficulty in unambiguously defining bioavailability makes it difficult to choose a method for measuring this parameter. Among the basic procedures is the measurement of toxicity to microorganisms. However, the results obtained allow an assessment of the cumulative effect of the test substance on microorganisms but do not indicate to what extent toxicity is due to the sensitivity of the microorganism concerned or to the bioavailability of the organic compound in question. Harmsen [17] noted that the measurement of biodegradability must take into account both chemical and biological aspects. Only such a complex approach allows an adequate risk assessment. 

Moreover, Semple et al. [12] compiled different methods to assess the degree of interaction of organic compounds with environmental microorganisms by identifying extractable organic substances. A similar approach was followed in the work of Riding et al. [18], who described methods using chemical oxidation in addition to extraction methods. They also drew attention to interfering factors that may interfere with bioavailability measurements. Particularly in environmental samples, interaction with small animals and worms becomes a problem. Another method may also be to determine changes in the populations of microorganisms exposed to a particular xenobiotic [19].

In summary, based on these mentioned definitions, and taking into account their possible limitations, a new, more general definition should be formulated to refer to the bioavailability of organic compounds to bacterial cells—especially when different fields of application are considered, e.g., antibiotic therapy, bioremediation, and other biotechnological processes. Hence, the bioavailability of a chemical compound (1) describes to what extent it can be assimilated by living cells, and (2) is the result of the transport of a chemical compound into the immediate vicinity of the cell (i.e., bioavailability at the physicochemical level) and the transport of the chemical compound into the cell through the cell wall and/or membrane (i.e., bioavailability at the cellular level). This proposed definition seems to be as wide as possible, and it is in this context that the term bioavailability is used in this study.

## 3. Significance of Bioavailability

The high bioavailability of organic compounds is crucial for cell function and can be considered to be both a positive and negative factor, both from the perspective of the microorganisms themselves and from that of humans. The uptake of chemical compounds that can be metabolised by microorganisms and used as a source of carbon and energy is undoubtedly positive for cells [20,21]. The opposite is true for the effects of substances that are toxic to them, leading to impairment of their vital functions and, sometimes, to cell death [16]. 

By regulating the bioavailability of chemical compounds, the condition of microorganisms is directly influenced [13]. This process is particularly important in three areas of human activity: biotechnological production using microbial cultures, bioremediation of organic pollutants, and the use of antibiotics and biocidal compounds to remove undesirable microorganisms—especially pathogenic ones (Figure 2).

### 3.1. Bioavailability in Biotechnological Processes

In biotechnological production, the bioavailability of the substrate(s) that are the sources of carbon, nitrogen, and energy for the cells is a fundamental issue [22]. If the substrate is cheap, it is possible to provide it in excess (e.g., saccharides in ethanol production), within the tolerance limits of the microorganisms [23]. However, even in this case, the amount of substrate available to the microorganisms may differ significantly from the theoretical values. For example, in the bioconversion of waste vegetable oils (to biosurfactants, biodiesel, etc.), the low solubility of lipids strongly affects the bioconversion efficiency [24].

The bioavailability issue also applies to organic compounds that are supplemented in microbial cultures. Vitamins, amino acids, and inhibitors or activators of metabolic pathways are added to the cultures at relatively low concentrations; however, being often relatively large molecules, their diffusion and membrane transport may be much slower than those of the main substrate [25,26]. In particular, during the logarithmic phase of cell population growth, their consumption is very high, and bioavailability may be the main factor slowing down this stage of biotechnological culture [27,28]. 

Low bioavailability may also be relevant in the bioconversion of compounds that are not the only substrate for microbial growth (e.g., in co-metabolic systems). Their low bioavailability may result in weaker competition with other compounds (with higher bioavailability) present in the culture and a decrease in bioconversion efficiency [29,30]. 

A separate issue is the bioavailability considered in the context of bioremediation of persistent organic pollutants (POPs), such as pharmaceuticals (including nitrofuran antibiotics), pesticides, and petroleum hydrocarbons. Most of the literature on their bioremediation focuses on the metabolism of these compounds, although many studies have indicated that it is the bioavailability that may in many cases determine the effectiveness of the biodegradation process [31,32,33,34].

POPs are xenobiotics, which either do not occur naturally in the environment or occur only in low concentrations, and are mostly poorly soluble compounds in aqueous solutions. In addition, they have a high affinity for solids, inorganic and organic soil particles, or the bottom sediments of water bodies [35]. This causes their limited desorption and, consequently, lower concentration in the aqueous phase. Liquid lipophilic compounds with a lower density than water also tend to form macroscopic layers on the water-phase surface. In contrast, solid particles of some contaminants form agglomerates or bottom deposits. In this case, intensive biodegradation can occur only on this small interface [31,32]. 

Another aspect is the low adsorption of contaminants on the cell surface, which limits their transport across the cell membrane. This assimilation step of organic compounds can also be affected by the size of the particles and the functional groups present in them. In numerous cases, these characteristics do not allow compounds to penetrate cells, because the permeability of the cell membrane is too low for them [36,37]. Consequently, the access of these compounds to microorganisms is limited and inefficient, which slows down the biodegradation process, even when biodegradation is not restricted by the metabolic capacity of the cells [38,39]. 

At this point, it should be noted that in situations of low bioavailability of a substrate present in a separate (e.g., oil) phase, some bacterial strains (e.g., of the genera *Bacillus* or *Pseudomonas*) produce biosurfactants [40]. These amphiphilic molecules—usually sugar–lipid or glycoprotein molecules—are very good emulsifiers, acting effectively even at relatively low concentrations. Their presence significantly increases the bioavailability of the substrate to the cells [32]. Biosurfactants can also modify the cell surface properties (e.g., by surface removal of lipopolysaccharides) and, thus, affect the bacteria’s hydrophobicity and substrate uptake [41].

In bacterial-culture-based biotechnological production and bioremediation processes, the possible high bioavailability of some compounds present in the microbial growth medium can also be a problem. In addition to the aforementioned problem that more bioavailable compounds may effectively reduce the metabolism of the desired substrate, the possible toxic effects of compounds with excessively high bioavailability should also be highlighted [20].

For most organic compounds, there is a certain limiting concentration above which a compound becomes toxic to a given bacterial strain. This is especially true for toxins with a biocidal effect (such as many of the POPs mentioned). However, negative phenomena may also be associated with those chemical compounds that are in principle safe for bacteria, such as commonly used substrates in biotechnological production [42]. For example, they may disturb the osmotic and electrolytic balance of the cell, excessively liquefy the cell membrane, or accumulate inside the cell. It is also possible for substrate inhibition to occur, slowing down the metabolism of a given substrate in a situation of excess substrate [43].

### 3.2. Bioavailability in Antimicrobial Therapies

Bioavailability is also one of the key phenomena affecting the effectiveness of antibiotics on bacterial cells. The antibacterial effect of antibiotics can be either bacteriostatic (i.e., inhibiting cell growth and proliferation) or biocidal (i.e., leading to cell damage and death). Antibiotics can be directed at blocking cell wall synthesis (such as beta-lactams), disrupting cell membrane function (e.g., polymyxins), impairing protein biosynthesis (such as aminoglycosides), or halting DNA replication (e.g., fluoroquinolones or nitrofurans) [44,45].

Every mechanism of action requires direct contact between the cell and the antibiotic molecule, and then the molecule has to penetrate the microbial wall [46]. However, if the antibiotic acts on the cell wall or membrane, it is not necessary for the antibiotic to penetrate completely into the cell [47]. 

Low bioavailability (to bacterial cells as well as to organisms infected by pathogens) is one of the factors considered to have a decisive impact on the performance of antibiotics—both synthetic and natural [48,49]. Furthermore, due to the non-specific toxicity of many bactericidal compounds, the antibiotic must be selected in such a way that it has a high affinity for the pathogen’s cells but a low affinity for the organism’s cells [50,51].

Bioavailability is also seen as a key issue in overcoming the problems caused by the spread of antibiotic-resistant strains. An increase in bioavailability (both to bacterial cells and organisms infected by pathogens) allows for increasing antibiotics’ effectiveness and maintaining their dose, which helps to fight infections caused by antibiotic-resistant strains [52]. On the other hand, the decrease in the bioavailability of an antibiotic in the environment may reduce the risk of antibiotic resistance gene expression in bacterial cells, as noted by Chen et al. [53]. 

## 4. Factors Influencing Bioavailability

Describing the phenomenon of bioavailability and briefly presenting its role in human-relevant processes, several issues affecting bioavailability have already been indicated. Among others, low water solubility and limited transport across the cell membrane have been mentioned. However, there are more factors influencing bioavailability, and they are divided into factors at the physicochemical level (i.e., resulting from the characteristics of the aqueous phase) and factors at the microbiological level (i.e., resulting from the characteristics of the living cell) [32,54,55]. 

Many physicochemical phenomena determine the bioavailability of organic compounds, and for this it is convenient to divide them into two subcategories: firstly, into processes determining the solubility of a given compound in the aqueous phase, which is essential for the growth of virtually all microorganisms [56]; and secondly, into processes related to the transport of molecules of these compounds in the aqueous phase [57] (Figure 3). Moreover, processes at the microbial level, in turn, can be divided into those related to the adhesion of the cell and the molecules of the chemical compound and those related to transport across the cell wall and membrane [16,58] (Figure 4).

### 4.1. Bioavailability at the Physicochemical Level

Chemical structure

Bioavailability at the physicochemical level is most influenced by phenomena that determine the solubility of a given organic compound in the aqueous solution in which the microorganisms live [59]. The solubility of a chemical compound results from its ability to be solubilised i.e., be solvated by water molecules (so-called hydration). It is favourably influenced by the ability to form hydrogen bonds between water molecules and functional groups such as hydroxyl (-OH), aldehyde (-CHO), carbonyl (=CO) and, to a lesser extent, amine (-NH_2_) and thiol (-SH) [59,60]. The absence of these listed functional groups results in very low solubility, usually not exceeding the order of ppm. Furthermore, the molecular size also affects bioavailability—smaller molecules are considered to be more bioavailable [61]. 

The chemical structure of organic compounds—in particular the functional groups—is responsible for the interaction of their molecules with their surroundings and, therefore, determines their bioavailability at a fundamental level. The phenomena influenced by the chemical structure are described in more detail below.

pH of the surrounding environment

The next factor positively influencing the solubility of a molecule is its susceptibility to ionic dissociation [62]. As a rule, compounds occurring as cations and anions show several orders of magnitude higher solubility. For example, propylparaben’s solubility at pH 9.5 is 25 times higher than at pH 7.5, and quetiapine’s solubility at pH 2 is 1000 times higher than at pH 8 [63]. Therefore, the presence of amine, carboxyl, phosphate, or sulfonic groups indicates potentially higher solubility of a given compound than its analogues without these groups [61,64]. 

However, it should be emphasised that organic compounds are for the most part weak acids and bases. Therefore, their dissociation in aqueous solutions strongly depends on the pH of the aqueous solution [62,63]. In the case of weak acids, solubility increases when they change from neutral to anionic form, which happens when the environment becomes more alkaline. In contrast, the concentration of the cationic form of weak bases increases when the pH decreases. This phenomenon is important in the case of orally administered pharmaceuticals, due to changes in pH in different sections of the gastrointestinal tract [65]. Moreover, this phenomenon is observed in the case of the bioavailability of environmental pollutants, including antibiotics such as nitrofurans [66,67].

Ionic strength and co-dissolved compounds

In aqueous solutions containing other solutes, the organic compounds compete with them for being hydrated. In the presence of a compound with higher solubility, the equilibrium between the dissolved and undissolved forms of the less soluble compound shifts towards the undissolved form; however, the interactions between compounds in one solution can be more complex [68]. This is particularly observed in the phenomenon of desalting when an increase in the ionic strength of the solution leads to a decrease in the solubility of organic compounds [66]. This phenomenon is used to regulate the solubility of proteins, among other things [69]. 

Moreover, the cations and anions present in the bacteria’s surroundings may affect the bioavailability. Divalent and multivalent ions tend to form complexes with organic compound molecules—e.g., with tetracycline—forming structures that have different solubility, diffusion and solubility parameters in solutions [70]. Ionic strength has a very important influence on the charge level of organic compound particles (e.g., their zeta potential), which is particularly important if their transfer takes place not in fully dissolved form but as nanoscale agglomerates [71].

Crystallinity and amorphousness

An important factor affecting the solubility of organic compounds is their existence in crystalline or amorphous forms. In the former case, if additional energy is required to break the bonds of the crystal lattice, the crystalline form is less soluble. Therefore, the amorphous form has a beneficial effect on the solubility of most compounds [60]. Moreover, polymorphs (i.e., anhydrous and solvate/hydrate forms) are recognised as improving bioavailability [72]. However, it should be remembered that if the amorphous form is not thermodynamically stable, it will gradually crystallise. Consequently, the solubility of amorphous compounds may decrease over time. Nevertheless, in some cases, the reverse process can occur [73]. 

The dissolution rate (although not the equilibrium concentration in solution) is also affected by the interfacial area between the chemical compound being dissolved and the aqueous solution [74]. The smaller the particles or droplets, the greater the mass exchange surface area [75]. Hence, colloidal systems of solid particles and liquid droplets (emulsions) are preferred to increase bioavailability [76]. The aim is to keep the particle/droplet size as small as possible, which is why micro/nanocolloids and micro/nanoemulsions are of such great interest [77,78].

Stabilisers and carriers

The formation of dispersed systems requires a relatively large expenditure of energy to disperse the organic compound of which the bioavailability is to be increased. Moreover, such systems are frequently unstable. Then, it can be observed that the solid particles they contain tend to aggregate and the liquid droplets tend to coalesce. For this reason, additional chemical compounds are used as stabilisers [79,80].

The function of stabilisers is usually performed by compounds of an amphiphilic nature, i.e., surfactants. They lower the surface tension energy and can change the surface charge of the droplet/particle in the dispersed system [81,82,83]. In addition, the nature of the interactions between the dispersed particles can be modified (e.g., from hydrophobic interactions to ionic ones) [84]. Thus, the effect of surfactants is both to facilitate the dispersion of the hydrophobic phase in an aqueous solution and to increase the thermodynamic stability of the system, increasing the chances of mass transfer between the particle/droplet and the solution and, subsequently, the bacterial cell. Surfactants in sufficiently high concentrations can also form micellar systems in which molecules of difficult-to-solubilise (and, thus, poorly bioavailable) compounds are encapsulated [32,85,86]. 

Another group of stabilisers used are solid particles that adsorb onto the interfacial surface and also lead to the stabilisation of dispersed particles or droplets. Pickering emulsions, in which nanoparticles are often used, are an example of such systems [87,88]. It should also be noted that an additional function of stabilisers of colloidal systems (both particulates and surfactants) is to modify the surface of the droplet/particle in such a way that it will have an increased affinity for the surface of the bacterial cell [89,90]. Adsorption of the droplets/particles directly onto the bacterial cell is then observed, which greatly facilitates the transport of the compound into the cell [32]. The function of the carrier can also be to facilitate a meeting between the bacteria and the organic compound if it has a high affinity for both [91]. Various drug delivery systems are also based on this mechanism, in which the carriers can be lipids (e.g., liposomes) as well as polymeric compounds—both natural and synthetic. Solutions are also used, in which the compound to be delivered to the cell is not encapsulated in the carrier but deposited on its surface. This approach includes many nanoparticle-based pharmaceutical delivery systems [92]. Among the compounds that increase bioavailability—especially antibiotics with relatively small molecules—polymers are very useful. They make it possible to create a variety of drug delivery systems, e.g., polymeric liposomes or micelles, highly branched polymers and dendrimers, and polymeric nanogels [93,94]. 

A separate group of compounds modifying the bioavailability of organic compounds to bacteria are ligands, such as organic acids with more than one carboxylic group. The enhanced bacterial uptake of antibiotic resistance response is positively related to the strength of organic ligands forming complexes with divalent metal cations [70]. 

However, the use of emulsion stabilisers and carriers is connected with the risk of not increasing bioavailability but decreasing it. Excessively strong binding of the delivered organic compound to the carrier or excessively high stability in the dispersed system can lead to reduced release into solution and reduced likelihood of bacterial contact with the molecule. This process was observed for some of the antibiotic carriers tested and for surfactant-assisted biodegradation of POPs, where biodegraded compounds became permanently entrapped in the micelles [95,96]. Furthermore, additional stabilisers and carriers introduced into the system can have undesirable effects on the cells—e.g., toxicity to cells cultured for industrial purposes—or, conversely, provide a medium for pathogenic microorganisms when an antibiotic delivery system is used [97,98].

### 4.2. Bioavailability at the Microbiological Level

Adsorption on cell surfaces

The surface of a bacterial cell has many functional groups that allow compounds present in the environment to adsorb onto it. The mechanisms of adsorption on living cells generally do not differ from those describing adsorption on non-living surfaces [99]. Thus, physical adsorption based on electrostatic bonds, hydrogen bonds, and hydrophobic interactions may occur. Chemical adsorption is also possible, involving the formation of chemical bonds between the adsorbed organic compound and the molecules of compounds that make up the outer layers of the cell [100]. However, in most cases, physical adsorption occurs [101].

As the outer layers of bacterial cells are dominated by bio-organic compounds with hydrophilic groups (mainly -OH), the cell surface exhibits strongly hydrophilic properties and a relatively high surface charge, which can be indirectly characterised by the zeta potential [32,102,103]. However, the properties of the cell surface undergo dynamic changes as the composition and structure of these layers change [104]. These changes occur as a result of cell growth and ageing, and also in response to changes in the external environment [105]. As a result, when describing the surface properties of bacterial cells, we can only characterise some of their average features. 

However, it is possible to intentionally influence the surface properties of bacterial cells so that, by decreasing or increasing adsorption, the bioavailability of a given chemical compound can also be regulated. For example, a pre-culture containing a hydrophobic carbon source will favour the proliferation of cells with specific hydrophobicity [106]. It is also possible to use amphiphilic compounds (e.g., surfactants) that, when adsorbed on the cell surface, will change the properties of the cell to the opposite, e.g., when the hydrophilic groups on the outer layer of the cell are attached to the hydrophilic groups of the surfactant, its hydrophobic groups will be exposed on the outside; the cell will then become more hydrophobic [32,107].

A separate issue is adsorption associated with the limited sorption of the organic compound in the biofilm formed outside the cell. The cell’s production of extracellular polymeric substances (EPSs)—mainly polysaccharides—is one of the more widely described cell defence mechanisms observed in both antibiotic and biodegradation processes [108,109]. This process leads to a significant decrease in the bioavailability and uptake of substances from the environment [110]. This is particularly true for high-molecular-weight compounds and those with hydrophobic properties. However, in certain cases, biofilm formation with EPSs facilitates the adsorption of organic chemicals such as antibiotics [111].

Another interesting aspect is the competition between different organic compounds for adsorption on the surface of bacterial cells. This applies, for example, to humic acids, which have a high affinity for bacterial cells, making it difficult for other compounds—such as tetracycline—to reach them [53]. Moreover, the bacteria and chemical compounds may co-adsorb on organic or inorganic particles, increasing the probability of contact between cells and organic compounds [91].

However, when considering adhesion processes at the cell surface, it is important to bear in mind that this process is usually far from equilibrium. As already mentioned, the cell surface is constantly undergoing changes, which may or may not affect the adhesive properties of the cell. More importantly, however, the adsorbed organic compound will to some extent be continuously taken up by and transported into the cell [112]. This makes it even more difficult to describe the process of adsorption of organic compounds onto the surface of a bacterial cell [113]. Nevertheless, a greater amount of adsorbed compound indeed favours greater transfer across the cell membrane [112]. 

Cell wall and membrane permeability

The vast majority of the metabolic processes of a living cell take place inside the cell, separated from the environment by a cell membrane and/or cell wall. In the case of very large molecules, the cell can produce extracellular enzymes that start the metabolic pathway outside the cell by breaking down complex compounds into simpler ones. These simpler compounds are then transported into the cell [114,115]. 

This transport into and out of the cell can be passive (i.e., resulting from purely physical phenomena) or active (i.e., based on the function of transport proteins in the membrane). It can be distinguished from the transport through specialised pores and channels in various membrane transporter proteins [116]. Hua and Wang [117] described three main mechanisms of substrate transport across bacterial membranes: (1) passive diffusion, (2) passive facilitated diffusion, and (3) energy-dependent/active uptake. In the case of organic compounds, active transport frequently plays a dominant role [117]. Saier et al. [118] classified membrane-active transporters of prokaryotic cells into channel proteins (i.e., transport via an energy-independent facilitated diffusion mechanism through a transmembrane pore), primary transporters (i.e., active transport that is coupled with ATP hydrolysis), and secondary transporters (i.e., active transport that is coupled with an electrochemical gradient) [118]. 

The number of transport proteins, their throughput, and the presence of coenzymes and cofactors strongly determine the efficiency of the transport process into the cell [119]. Blocking their action impairs cell function precisely because of the reduction in nutrient bioavailability. Moreover, the deactivation of the functions of membrane transporters decreases the bioavailability of a wide range of organic compounds [120]. 

The physical permeability of the cell membrane (i.e., its fluidity and porosity) changes as a result of exposure to a changing external environment [121]. A decrease in membrane permeability may occur as a result of the cell’s defence response to the presence of toxins. It may manifest as a change in the profile of membrane fatty acids, among which saturated acids begin to dominate in place of unsaturated ones [122,123]. 

As the cell membrane is made up of phospholipids, it is sensitive to amphiphilic compounds such as surfactants. Contact with them will usually lead to an increase in the permeability of the cell membrane [39]. It will also be affected by hydrophobic compounds, which have a relatively high affinity for membrane lipids. In this case, the effect on the resultant permeability is ambivalent; depending on their chemical structure, they may increase or decrease its fluidity. Increased fluidity favours the physical diffusion of organic compounds across the cell membrane and, consequently, increases their bioavailability [37,124]. Considering affinity for transporting proteins, O’Shea and Moser [115] stated that antibiotics—especially those targeting Gram-negative bacteria—are statistically more hydrophilic than other drugs targeting higher organisms’ cells [125]. Furthermore, the presence of divalent cations may also limit the permeation of organic compounds across biological membranes, due to the formation of complexes by these ions with these compounds [70].

The impact of osmotic stress on chemicals’ uptake is an additional aspect influencing bioavailability at the microbiological level [126]. Although osmotic stress is perceived to be unfavourable for the transport of compounds into the cell, the water flux between bacterial cells and the surrounding high-salinity water phase reduces the bioavailability of dissolved chemicals [127]. However, this relationship is not unambiguous, as osmotic stress promotes greater membrane permeability as the cell seeks to equalise osmotic pressure. As was noted by Chen et al. [128], enhanced antibiotic bioavailability to bacteria could result from compromised cell membranes and enhanced membrane permeability in hypertonic solutions.

Positive and negative chemotaxis

Chemotaxis should be mentioned as the last (but not least) of the main factors determining bioavailability [129]. It applies only to cells capable of independent motility. This ability is used by bacterial cells to move towards a more favourable environment [130]. Using receptor proteins, cells can recognise the presence of a chemical compound [131,132]. The cell can move toward attractant chemicals or away from repellents. However, this movement is related to the recognition of changes in concentration and, therefore, occurs along the concentration gradient vector [133].

This ability is specific and genetically determined. A change in chemotaxis can occur as a result of genetic mutations, leading to strains lacking chemotaxis [134]. However, it is not possible to significantly influence this factor by changing environmental conditions. However, if chemotaxis is a negative factor and should be minimised, it is possible to provide homogeneity of the used organic compound in the solution surrounding the cell (i.e., no concentration gradient), or to use strains with a mutation that deprives the cell of chemotaxis [135]. Conversely, when higher bioavailability is desired, working with a strain that exhibits chemotaxis is advantageous [40,131,136]. However, the strain selection option only applies to selected cases (e.g., biotechnological processes) and is not applicable in cases where strain selection is not possible (e.g., antibiotic therapy). Moreover, for specific bacterial strains, some organic compounds (e.g., phenol) can be attractors at low concentrations but repellent (i.e., causing negative chemotaxis) at higher concentrations [137].

## 5. Summary

As presented in the above work, the bioavailability of organic compounds to bacterial cells is a very important issue that is crucial in two areas of human activity in particular: in biotechnological processes based on bacterial cultures, and in the development of antibacterial therapies based on antibiotics. 

Among the range of factors that affect bioavailability are those involving physicochemical processes in the solution in which the bacterial cells function and those that relate directly to the properties and life processes of the cells. Increased solubility and, therefore, bioavailability is favoured by the presence of the compound in hydrated and ionic forms. It is also beneficial to increase the mass exchange surface area, i.e., to disperse the undissolved organic compound as much as possible, which occurs in colloidal and emulsion systems. Stabilisers (e.g., surfactants) and carriers used for this purpose can further modify the surface area of particles/droplets by increasing their direct affinity for cells. However, they may themselves negatively interfere with the interaction between the dispersed organic compound and the bacterial cell.

Key factors influencing bioavailability at the microbial level are the adhesion of the organic compound to the cell surface and the permeability of the cell membrane. The higher these parameters are, the higher the observed bioavailability. An additional factor favouring bioavailability is cell chemotaxis, i.e., active movement towards the chemical compound in question. 

Regardless of whether increased bioavailability is a desirable or undesirable phenomenon in a given case, knowing how many factors affect it—and in which ways—allows us to regulate it effectively, allowing for greater efficiency and effectiveness of our actions involving bacterial cells.

## Figures and Tables

**Figure 1 molecules-27-06579-f001:**
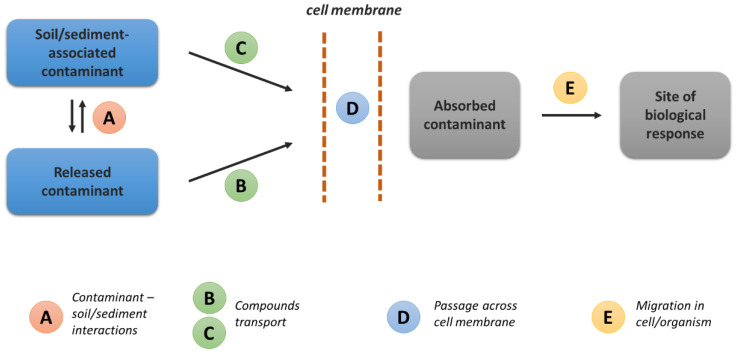
Bioavailability of organic chemicals according to Ortega-Calvo et al. [13]. Adapted with permission from Ref. [13]. Copyright 2013, American Chemical Society.

**Figure 2 molecules-27-06579-f002:**
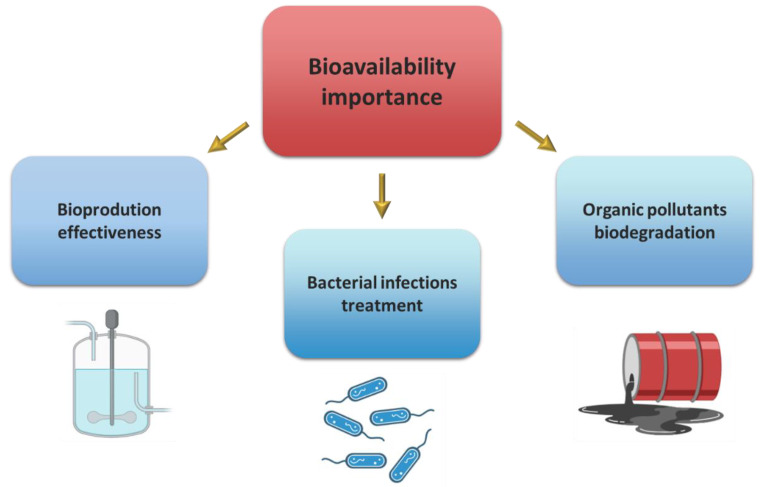
Main areas of importance of organic chemicals’ bioavailability to bacteria cells.

**Figure 3 molecules-27-06579-f003:**
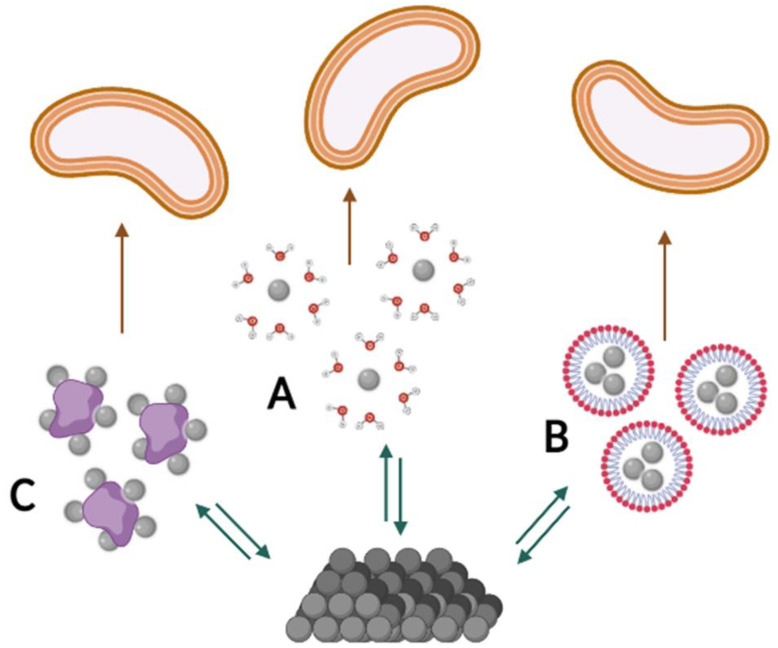
Main strategies of increasing of bioavailability of organic compounds (grey circles) at the physicochemical level: (**A**) solubilisation; (**B**) transport in micelles; (**C**) transport on carriers.

**Figure 4 molecules-27-06579-f004:**
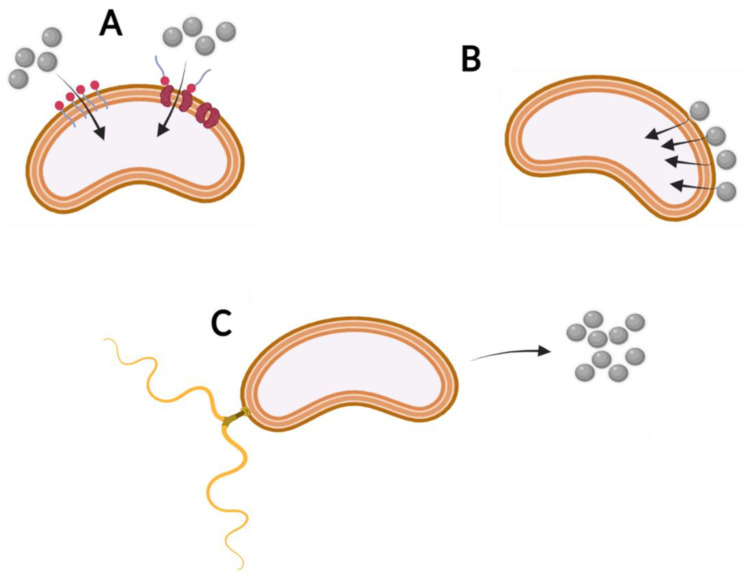
Main strategies of increasing bioavailability of organic compounds (grey circles) at the microbiological level: (**A**) cell membrane modification, e.g., with surfactant; (**B**) organic compound adsorption on the cell surface; (**C**) chemotaxis towards chemicals.

## Data Availability

Not applicable.

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
