# Peer review of "Factors Influencing the Bioavailability of Organic Molecules to Bacterial Cells—A Mini-Review"

_molecules, 2022, doi:10.3390/molecules27196579_

Round 1
Reviewer 1 Report
The manuscript entitled "Factors influencing the bioavailability of organic molecules to bacterial cells – a mini-review” by Smułek and co-workers is nicely written and has scientifically presented. The development of anti-bacterial medicines based on antibiotics and biotechnological processes based on bacterial cultures are two areas of human activity where the bioavailability of organic chemicals to bacterial cells is particularly relevant.
This manuscript suggests a taxonomy of variables affecting bioavailability, separating them into physicochemical factors and microbiological factors. Each of the elements listed may be used to alter bioavailability in the desired manner by being aware of their significance and the processes influencing them.
1. Authors should discuss the measurement of bioavailability.
2. I recommend that the authors keep and structure the manuscript point by point, such as the effect of pH, the effect of carrier protein, water and lipid solubility, molecular size, stability, H-bonding, membrane physiology, and so on.
3.Authors should discuss various approaches for the enhancement of drug bioavailability.
4. The main content of this manuscript is discussed under "3. Factors influencing bioavailability." It is too short for a review article. I am suggesting the authors elaborate and discuss thoroughly, and also incorporate some new points to enrich the manuscript. Thus, I am recommending this manuscript for major revision.
Author Response
Response to Reviewers Comments
Dear Reviewers,
We thank the Reviewers for their time spent carefully reviewing the manuscript, and in their opinions regarding the science and presentation of the material. We made sure that each one of the Reviewers’ comments has been addressed carefully and the paper is revised accordingly.
Please find below the answers to your valuable comments:
Reviewer #1
The manuscript entitled "Factors influencing the bioavailability of organic molecules to bacterial cells – a mini-review” by Smułek and co-workers is nicely written and has scientifically presented. The development of anti-bacterial medicines based on antibiotics and biotechnological processes based on bacterial cultures are two areas of human activity where the bioavailability of organic chemicals to bacterial cells is particularly relevant.
This manuscript suggests a taxonomy of variables affecting bioavailability, separating them into physicochemical factors and microbiological factors. Each of the elements listed may be used to alter bioavailability in the desired manner by being aware of their significance and the processes influencing them.
- Authors should discuss the measurement of bioavailability.
Answer: Indeed, this issue has not been addressed by us and we thank Reviewer for bringing it to our attention. We have expanded the relevant section in the introduction, which reads as follows:
“The difficulty in defining bioavailability unambiguously makes it difficult to choose a method for measuring this parameter. Among the basic procedures is the measurement of toxicity to microorganisms. However, the results obtained allow an assessment of the cumulative effect of the test substance on microorganisms, but do not indicate to what ex-tent toxicity is due to the sensitivity of the microorganism concerned and to what extent to the bioavailability of the organic compound in question. Harmsen [17] points out that the measurement of biodegradability must take into account both chemical and biological aspects. Only such a complex approach allows an adequate risk assessment.
Moreover, Semple et al. [18] compiled different methods to assess the degree of interaction of organic compounds with environmental microorganisms by determining ex-tractable organic substances. A similar approach was followed in their work by Riding et al . [19] describing methods using chemical oxidation in addition to extraction methods. They also drew attention to interfering factors that may interfere with bioavailability measurements. Particularly in environmental samples, interaction with small animals and worms becomes a problem. Another method may also be to determine changes in the population of micro-organisms exposed to a particular xenobiotic [20] .”
- Harmsen, J. Measuring Bioavailability: From a Scientific Approach to Standard Methods. J. Environ. Qual. 2007, 36, 1420–1428, doi:10.2134/jeq2006.0492.
- Semple, Kirk.T.; Doick, K.J.; Jones, K.C.; Burauel, P.; Craven, A.; Harms, H. Peer Reviewed: Defining Bioavailability and Bioaccessibility of Contaminated Soil and Sediment Is Complicated. Environ. Sci. Technol. 2004, 38, 228A-231A, doi:10.1021/es040548w.
- Riding, M.J.; Doick, K.J.; Martin, F.L.; Jones, K.C.; Semple, K.T. Chemical Measures of Bioavailability/Bioaccessibility of PAHs in Soil: Fundamentals to Application. Journal of Hazardous Materials 2013, 261, 687–700, doi:10.1016/j.jhazmat.2013.03.033.
- Lindgren, J.F.; Hassellöv, I.-M.; Dahllöf, I. PAH Effects on Meio- and Microbial Benthic Communities Strongly Depend on Bioavailability. Aquatic Toxicology 2014, 146, 230–238, doi:10.1016/j.aquatox.2013.11.013.
- I recommend that the authors keep and structure the manuscript point by point, such as the effect of pH, the effect of carrier protein, water and lipid solubility, molecular size, stability, H-bonding, membrane physiology, and so on.
Answer: We completely agree with this valuable Reviewer's comment. The main body of the paper, section 4, has been reorganized and completed, so that the text is now clearer and presents a broader perspective of the issues described.
- Authors should discuss various approaches for the enhancement of drug bioavailability.
Answer: Indeed, this topic has not been sufficiently developed. However, the literature tends to focus on the bioavailability of drugs to the human body (possibly animals or plants) rather than to the cells of microorganisms. We have nevertheless supplemented the literature on the subject and added additional information to our manuscript in section 4.
- The main content of this manuscript is discussed under "3. Factors influencing bioavailability." It is too short for a review article. I am suggesting the authors elaborate and discuss thoroughly, and also incorporate some new points to enrich the manuscript. Thus, I am recommending this manuscript for major revision.
Answer: We totally agree with this comment. We developed the section 4. “Factors influencing bioavailability”, having in mind to be possibly coherent and concise, due to the fact that our article was planned as a mini-review.
Reviewer 2 Report
The current review presents the topic of bioavailability of organic compounds (nutrients, vitamins, toxins, antibiotics, pollutants etc) to bacterial cells from the chemical and microbial cell perspective.
The topic is adequately presented, though it could be a bit shorter in the beggining.
L390-394 The two sentences are repeating themselves
L329 change "consciously" into "willingly" or similar
Author Response
Response to Reviewers Comments
Dear Reviewers,
We thank the Reviewers for their time spent carefully reviewing the manuscript, and in their opinions regarding the science and presentation of the material. We made sure that each one of the Reviewers’ comments has been addressed carefully and the paper is revised accordingly.
Please find below the answers to your valuable comments:
Reviewer #2
The current review presents the topic of bioavailability of organic compounds (nutrients, vitamins, toxins, antibiotics, pollutants etc) to bacterial cells from the chemical and microbial cell perspective.
- The topic is adequately presented, though it could be a bit shorter in the beginning.
Answer: Thank you for this valuable comment. We agree that in its original form the introduction was too long. We have decided to extract from it a separate chapter that focuses on the definition of bioavailability. In this way, the current introduction focuses more on the background and reasons of our study, which we believe will be clearer to the reader.
- L390-394 The two sentences are repeating themselves
Answer: Thank you for the comment, the mentioned paragraph is now as follows:
“Considering affinity to transporting proteins, O’Shea and Moser stated, that the antibiotics, especially those targeting Gram-negative bacteria, are statistically more hydrophilic than other drugs targeting higher organisms cells. However, there is a noticeable difference between Gram-negative and Gram-positive bacteria targeted antibiotics – the latter are more hydrophobic.”
- L329 change "consciously" into "willingly" or similar
Answer: We changed the sentence as the Reviewer rightly commented. Its current version is enclosed below:
“However, it is possible to influence willingly the surface properties of bacterial cells so that, by decreasing or increasing adsorption, the bioavailability of a given chemical compound can also be regulated.”
Round 2
Reviewer 1 Report
I am fully satisfied with the revised manuscript. The authors did extensive revisions according to my suggestions. Now this revised manuscript is suitable for publication. Therefore, I am recommending this revised manuscript for acceptance for publication.